# Ingestion of Soil by Grazing Sport Horses

**DOI:** 10.3390/ani11072109

**Published:** 2021-07-15

**Authors:** Stefan Jurjanz, Claire Collas, Carol Quish, Bridget Younge, Cyril Feidt

**Affiliations:** 1UR Animal et Fonctionnalités des Produits Animaux (URAFPA), Université de Lorraine—INRAE, F-54000 Nancy, France; Claire.Collas@univ-lorraine.fr (C.C.); Cyril.Feidt@univ-lorraine.fr (C.F.); 2Department of Biological Sciences, Faculty of Science and Engineering, University of Limerick, V94 T9PX Limerick, Ireland; carol.quish@ul.ie (C.Q.); bridget.younge@ul.ie (B.Y.)

**Keywords:** soil ingestion, equids, herbage offer, health, pasture, welfare

## Abstract

**Simple Summary:**

Soil ingestion has been well documented for the majority of outside reared animals but not in horses. As soil can be a vector of environmental pollutants, such studies generally aim at controlling exposure to pollutant uptake in food producing animals. In horses, ingestion of soil may cause gastrointestinal disorders such as sand colic or intestinal damage. Therefore, soil ingestion has been studied in Irish sport horses grazing at three levels of herbage offer: 2, 3 or 4% of their body weight. Their soil intake was around 4% of the totally ingested dry matter corresponding to 543 to 648 g of dry soil per animal per day, which is quite similar to cattle in normal grazing conditions. Such amounts would clearly be an issue for food safety in areas with contaminated soil but also an animal welfare issue due to gastrointestinal damage. The height of the pastured grass sward seems to be a reliable criterion to indicate the level of risk of soil intake when horses graze short herbage in close proximity to the ground surface and should be moved to a new paddock.

**Abstract:**

Data on soil ingestion in horses are lacking in contrast to other free-range animals. The importance of soil as a vector for environmental pollutants to food is less relevant in horses but several disorders secondary to soil ingestion, such as sand colic or enteritis have been reported. Therefore, soil ingestion has been studied on Irish sport horses grazing at three offered levels of daily herbage: 2, 3 and 4% of their body weight. Soil ingestion was estimated by the faecal recovery of a soil natural marker. Horses had 4.5, 4.1 and 3.7% of soil in their total intake respectively for the 2, 3 and 4% herbage offers. The 4% offer presented significantly less intake (543 g/d) compared to the more restricted offers (624 and 648 g respectively for 3 and 2%). The post-grazing sward height was significantly lower on the 2% offer (3.1 cm) compared to the higher offers (4.1 and 4.4 cm respectively for 3 and 4%). Thus, restricted herbage allowance made grazing closer to the ground and increased soil ingestion. The sward height appeared to be a reliable indicator to manage animal withdrawal from a pasture to limit soil ingestion and the risk of gastrointestinal pathologies caused by it.

## 1. Introduction

In general, soil ingestion has no nutritional purpose except counterbalancing significant mineral deficits described as geophagia [1]. Nevertheless, it has some negative consequences. Firstly, global digestibility of the feed decreases as soil is very poor in organic matter, and its presence in the digestive tract disturbs the digestive flora. However, the main risk is that soil may be a vector of environmental pollutants, which can accumulate over a very long time in soil and is therefore one of the principal exposure pathways to food producing animals [2]. Hence, the evaluation of soil ingestion is a central point in risk assessment approaches. It has been studied in numerous food producing animals such as dairy cows [3,4], beef cattle [5,6], sheep [7,8], free-range pigs [9], free-range hens [10] and chicken [11]. Nevertheless, no such data are available in horses. In this species, the frame of food safety evaluation would not really match since consumption of horsemeat is—except in Central Asia (e.g., China, Kyrgyzstan, and Kazakhstan)—very low. Even if this reduced issue of food safety can explain the lower interest of the scientific community to study soil ingestion in horses, there are pathologic concerns justifying investigations. Indeed, superficial soil can carry an important amount of microorganisms in the stomach. Its relatively high pH would not very efficiently limit microbial growth. In addition, a deposit of ingested soil may sensitively reduce the relatively small volume of the horse’s stomach. Moreover, Kilcoyne et al. [12] and also Niinstro et al. [13], both describe pathologic consequences such as sand colic or eroded intestinal mucosal lining without quantifying the responsible amount of ingested sand. Siwinska et al. [14] reported proportions of sand in stool but did not evaluate the corresponding level of ingestion. Thus, the ingestions of soil and sand by horses and its pathologic consequences has been clearly shown but no available data allows more accurate evaluation in horses.

Animals of comparable size, such as adult cattle, may ingest 30 to 100 g of dry soil per 100 kg of BW daily [3,6] under classical pasture conditions, and even up to 150 g per 100 kg of BW under poorer conditions [4,15], but the grass prehension differs notably between both species. Indeed, the height of the available grass is objectively measured with a herbometer and expressed as sward height (SH). It has been shown in ruminants that sward height (SH), especially post-grazing, is a good predictor of the level of soil ingestion [4,6]. Indeed, the shorter the grass sward, the closer to ground surface do animals graze and can consequentially ingest more soil along grass prehension. The anatomy of the equine mouth (i.e., movable lips and presence of maxillary) allows them to graze closer to the ground surface in comparison to cattle. This raises the following questions: Would the ingestion of soil by horses also increase when grass offered decreases, or would the difference of grass prehension between both species not (or less) expose horses to an increased risk of soil ingestion?

In the absence of any published data, the aim of this study was firstly to evaluate the ingestion of soil in realistic grazing conditions and secondly to investigate the link between the SH and level of soil ingestion in the equine species.

## 2. Materials and Methods

The study took place from 12 July to 2 September 2017 on a permanent grassland pasture located in Monaviddogue, Oola (Co Limerick Ireland). The experiment was conducted in farm like conditions and approved (Approval Number, 2017_6_1_ULAEC) by the University of Limerick Animal Ethics Committee (ULAEC).

### 2.1. Animals and Experimental Design

Six adult Irish Sport horses weighing 623 ± 32 kg (mean ± SEM) at the nutritional requirement level of maintenance and aged between 4 and 10 years were grouped by body weight (BW) in pairs, which followed the treatments in the same order.

Three levels of Daily Herbage Allowance (DHA) were studied in these grazing horses during three successive periods via a Latin Square Design 3 × 3. DHA was expressed depending on the BW. The first level of DHA corresponded to the grass amount, which needs to be ingested in order to meet their daily requirements [16], i.e., 2% of the BW. Then grazing losses by trampling or selective ingestion behavior were taken into account defining two DHA levels 1.5 and 2 times higher at 3% and 4% of the BW.

Each period lasted 16 days and consisted of 10 d of adaption of the horses to the DHA on the new pasture plot followed by 6 days carrying out the intake measurements.

### 2.2. Grazing Management and Vegetation Characteristics

The study was carried out on a 2.6 ha plot of permanent pasture grassland covered with a predominant perennial ryegrass sward. The plot was divided into three paddocks. One paddock was used for each period and was mown beforehand at a different date to ensure identical vegetation stage for measurements. Each paddock was divided into three subpaddocks for each of the three pairs of horses receiving one of the three DHAs. The horses were managed with strip-grazing through the sub-paddock to control the DHA by adjusting the area offered relative to the herbage mass (HM) available on the day prior to grazing. The available HM on the next strip to be grazed was estimated every two days by measuring the pre-grazing sward height (SH), and using a SH-HM regression established on the experimental plot.

SH and HM were simultaneously measured once weekly in 0.09 m^2^ squares (30 by 30 cm) randomly selected among areas of short, medium and tall swards on the next area to be grazed. SH inside each square was measured before and after herbage cut at 3.5 cm from the ground surface. For each square, herbage cut was weighed and then divided into two samples. One sample was dried for 24 h at 103 °C to determine the dry matter (DM) content of grass, then HM was calculated for the regression. The second samples from each square were pooled by sub-paddock to form a representative sample from each treatment at each measurement period. Representative pooled samples were dried during 48 h at 60 °C and crushed over a 2 mm sieve before the acid insoluble ash (AIA) analyses necessary to assess soil ingestion.

The strips allocated to each DHA level were made using temporary electric fencing, and the two horses assigned during one period to a given treatment moved every second day with measurements of pre and post-grazing SH conducted using a rising plate herbometer. Back fencing prevented access by horses to the previously grazed strip. This management enabled us to offer the animals an identical sward and DHA during each 16-day-period. At each new time period, animals were weighed and the amount of grass to be offered was calculated to achieve the new DHA level based on the horses’ BW.

### 2.3. Dry Matter Intake

The daily grass DM intake was estimated for each horse during the last 6 d of each period following the methodology described by Collas et al. [17] by dividing the faecal output (dry weight over 24 h, attributable to grass) by the indigestible proportion of ingested grass (i.e., 1—DM digestibility of ingested grass). Therefore, the faeces of all animals were individually collected and weighed each day of measurement. Each dropping pile on the plot was attributed to one horse using small coloured plastic balls (1 colour per horse), which the animals received via a daily supply of 200 g of rolled barley (160 g DM), ingested without refusals. The total faecal output was weighed on a daily basis during the measurement period for each horse. A representative sample was collected daily from the center of the dropping piles in order to avoid soil or dust contamination, then dried for 72 h at 60 °C to determine DM content, and afterwards crushed over a 2 mm sieve before AIA and crude protein (CP) analyses.

The DM digestibility of ingested grass was estimated via the faecal CP content attributable to grass according to the equation of Mesochina et al. [18]. A correction was applied to take into account the amount of faecal CP attributable to barley [19] from CP content of barley and from the apparent CP digestibility of barley [16]. Finally, the total ingested DM corresponded to the sum of the previously estimated grass intake and 160 g DM of rolled barley.

### 2.4. Ingestion of Soil

Soil ingestion rate (SIR) was estimated by the method of Beyer et al. [20] and validated by Jurjanz et al. [21] using an internal marker: acid insoluble ash (AIA). Indeed, such marker compounds have been much more concentrated in the fraction, which should be traced (i.e., soil) in comparison to other ingested matrices (i.e., feed) as shown in previous studies [4,9,11]. Using its concentrations and an estimate of the DM digestibility of the total diet (dDM), soil ingestion rate can be estimated as follows:SIR[%] = (AIA_feed_ − AIA_feces_ + dDM × AIA_feces_)/(dDM × AIA_feces_ − AIA_soil_ + AIA_feed_)(1)

Moreover, surface soil (layer 0–5 cm) was sampled from each plot at each period. Then, AIA concentrations were analyzed in samples of soil, feed (grass and barley) and faeces as described previously [4]. The determination of the DM digestibility of the total diet (dDM) was obtained from the following equation:dDM = 1 − (daily total fecal DM output/daily total DM intake)(2)

Quantity of ingested soil (QIS) was calculated by multiplication of SIR and total ingested feed (grass and barley).

### 2.5. Statistical Analyses

Statistical analyses were performed using R software, version 3.6.1 [22]. Intake variables (grass and total intakes, faecal output, DM digestibility), body weight and soil ingestion variables (fecal AIA content, SIR, QIS), were tested using Linear Mixed Models (LMM) with treatment, period and treatment × period interaction as fixed effects, and individual as a random effect. A non-significant interaction between treatment and period was removed from final models. Each horse represented an experimental unit. Significant effects were declared at *p* < 0.05 but tendencies (i.e., *p* < 0.10) were also indicated. Multiple comparisons for each significant factor were realized using Tukey correction (*p* < 0.05) with glht function of multcomp package.

## 3. Results

### 3.1. Grazing Parameters and Body Weight

The pre-grazing sward height was 11.9 cm without any difference between the three treatments (Table 1). Contrarily, after grazing, the sward height at the lowest DHA was 3.1 cm and was significantly lower than the 4.4 cm at the highest DHA. The post-grazing SH at the intermediate DHA (i.e., 3% of the BW) of 4.1 cm did not significantly differ from the other treatments (Table 1). Nevertheless, the post-grazing SH tended to be lower between the lowest and the two higher DHAs (3.1 cm for 2% vs. 4.1 cm for 3% and 4.4 cm for 4%, *p* < 0.10).

The horses had a lower body weight on the smallest DHA in comparison to the two higher DHAs, 610 and 629 kg, respectively (*p* < 0.05).

The AIA content of the allowed grass of 18 g/kg DM on average did not significantly differ between treatments. In soil, there was a slightly lower content of AIA in the plots pastured for the intermediate treatment (678 vs. 697 g/kg dry soil) but this very small difference did not reach the significance threshold (*p* = 0.07). Thus, the marker fraction AIA was 40 times more concentrated in soil than in the pastured grass (Table 1) allowing us to efficiently mark the soil.

### 3.2. Grass Intake and Faecal Output

The grass intake of the animals of 14.5 kg DM/d did not differ significantly between the treatments (Table 1) even if the raw mean between the lowest treatments differed by 1.2 kg/d (14.0 and 15.2 kg/d respectively for the DHAs of 2 and 3%, *p* > 0.10). The digestibility of the ingested grass did not differ between the three treatments with 0.47 on average (Table 1), but varied slightly between the second and the third period of the experiment (0.46 and 0.48 respectively, *p* < 0.05). In addition, the faecal output of 7.7 kg DM/d on average did not differ between treatments (Table 1). The AIA content in the faeces decreased significantly from the lowest to the highest DHA (from 60.3 to 51.1 g/kg DM respectively, *p* < 0.05) with an intermediate concentration of 55.3 g/kg DM (NS) for the DHA of 3% (Table 1). It is worth mentioning that horses during the first experimental period had significantly lower AIA contents in their faeces than during the periods 2 and 3 (respectively 47.9 and 59.4 g/kg DM, *p* < 0.01) but without any significant interaction with the treatment (Table 1).

### 3.3. Ingestion of Soil

The soil ingestion rate (SIR) increased significantly when the DHA decreased, from 3.7% at the highest DHA to 4.5% at the DHA of 2% (*p* < 0.05, Table 1). The horses had an intermediate SIR of 4.1% at a DHA of 3%, which did not significantly differ to the other treatments. In addition, the SIR was significantly (*p* < 0.05) lower during the first experimental period (3.3%) compared to the other two periods (4.5%) without any interaction to the treatment (Table 1).

The quantity of ingested soil (QIS) increased significantly (*p* < 0.05) from 543 g/d at the highest DHA to 636 g/d at the two more restrictive DHAs (Table 1). The slight numeric difference between the DHA treatments of 2 or 3% did not reach the significance threshold (*p* > 0.10). The relative soil ingestion per 100 kg of BW followed the same hierarchic order between the treatments (Table 1): highest for the lowest DHA at 107 g/100 kg BW and was significantly (*p* < 0.05) higher than for the highest DHA at only 89 g/100 kg BW. Horses had an intermediate relative QIS at the intermediate DHA at 99 g/100 kg BW, which did not differ significantly from the two other treatments.

Absolute and relative QIS are significantly affected by the experimental period (*p* < 0.001) but every time without any interaction with the treatment (Table 1). In both cases, QIS in the first experimental period was significantly lower (484 g/d and 79 g/100 kg BW respectively for absolute and relative QIS) than during the periods 2 and 3 (666 g/d and 108 g/100 kg BW respectively for the absolute and relative QIS).

## 4. Discussion

### 4.1. The Very First Quantification of Soil Ingestion in Horses

This very first quantification of soil ingestion by grazing equids showed proportions between 2.6 and 5.3% of the totally ingested DM, corresponding to 388 to 845 g per animal per day. Therefore, the ingestion of soil seems quite notable and should be integrated in risk evaluations for food safety and health. Indeed, such amounts could transport significant amounts of harmful compounds if the soil contained any cadmium or arsenic for example, but could also disturb the gastrointestinal milieu by its mechanical and microbiological adverse effects.

Such levels of soil ingestion correspond to what has been reported in other herbivorous animals grazing on sufficient herbage allowances, for instance dairy cows [3,4] or sheep [7,23]. Indeed, in such grazing conditions, these animals would hardly ingest more than 100 g of soil per 100 kg of BW corresponding to 10 g of soil per kg metabolic weight in line with our observations in grazing Irish sport horses. Nevertheless, the soil ingestion would significantly increase when pasture conditions deteriorate as winter grazing [8], in arid zones [15] or very high stocking rates [24]. In some extreme cases combining high stocking rates with very humid conditions, the proportion of soil in the ingested DM exceeds 20% during short periods [23,24,25]. Although these effects have been shown only in sheep and cattle, it seems likely that deteriorated grazing conditions would also increase the soil ingestion in horses, but the level of such increases needs to be studied.

### 4.2. DHA and SH Are Good Indictors to Limit the Ingestion of Soil

Pre-grazing SH (11.9 cm on average) did not differ significantly between treatments. As the post-grazing SH significantly decreased with DHA reduction (4.4 cm for DHA 4% vs. 3.1 cm for DHA 2%), indicating a restricted forage supply and by consequence inducing horses to graze shorter.

There were no significant differences in grass intake between treatments with an average of 14.6 kg DM per day (or 2.36% BW). Nevertheless, DHA were obtained by estimating the available herbage mass every 2 days by cutting the grass at 3.5 cm from the ground. Therefore, the grass of the 0–3.5 cm stratum was not taken into account in the estimates of DHA even though it is, at least partially, available to horses. On the one hand, horses are able to graze very close to the ground by means of the presence of both mandibular and maxillary incisors. On the other hand, the grass within the first few centimetres above the ground is more fibrous and has a higher DM content than grass in the leafy parts further from the ground. In dairy cows, Pérez-Prieto et al. [26] observed that the effect of pre-grazing grass mass on grass intake was affected by the height above which forage supply was estimated. According to the literature, both cattle and equines can graze down to 2–3 cm, even less for equines [17,26,27]. This underestimation of the amount of forage offered has allowed the horses on DHA 2% to achieve a daily grass intake of 14.0 kg DM (2.33% BW) to cover their requirements by grazing shorter (<3.5 cm) than horses on DHA 3 and 4% (>4 cm).

On the three treatments, the horses managed to ingest enough grass to cover their requirements. Although the horses grazed shorter on the DHA 2%, this did not influence the DM digestibility of the ingested grass (on average 0.47 whatever the DHA). The post-grazing SH differences between the three treatments did not influence the quality or quantity of ingested grass. However, a significantly higher soil ingestion was observed on the DHA 2% than on the DHA 4%, whether soil ingestion was expressed as a percentage of total DM intake, as a quantity per animal, or as a ratio of live or metabolic weight of the horses. On the DHA 2%, grazing closer to the ground may have resulted in a higher soil ingestion either through direct contact with the soil surface or through the intake of grass closer to the ground and therefore potentially more soiled by soil particles adhered to the vegetation (dust, splash effect during a rainy episode, trampling, etc.).

In a study testing three DHA levels (35, 52.5 and 70 g DM/kg BW/d, estimated at ground level) with lactating saddle mares in a similar strip grazing system (new strip of fresh herbage offered every two days), Collas et al. [17] reported a linear effect of DHA on grass DM intake with an increase of 0.13 kg DM ingested/kg DM of grass offered at ground level. These authors also observed a DHA influence on grazing behaviour with daily grazing time significantly longer on medium and high than on low DHA. Grazing activity on the second day on a strip was mainly influenced by DHA. Thus, it would be interesting to study grass and soil intakes of horses under more restrictive grazing conditions than those in the present study (e.g., DHA estimated at ground level). Such conditions could result in lower grass intake and higher soil ingestion (as a % of total intake) than reported in the present study, at least for the lower DHA, with significant differences between treatments.

### 4.3. Consequences on Health

Ingestion of soil by horses has often been reported, especially with deleterious effects on horses’ health justifying better integration of this aspect in the evaluation of animal welfare.

Firstly, different authors reported an uptake of consequent amounts of metals inducing toxic effects, which can hardly be explained only by the metal content in grazed vegetation. Indeed, Eamens et al. [28] reported zinc toxicity and hypocuprosis near industrial plants and others [29] also reported zinc poisoning with hypocuprosis in a mining district. Even though grass is often the main pathway of zinc ingestion, Madden [29] attributed an increase of zinc exposure to a stunted vegetation, implying a low herbage allowance and increased soil ingestion. Nevertheless, the most common symptoms related to soil ingestions by horses are attributable to sand accumulation and sedimentation in the intestine, with sand colic, diarrhea and enteritis developing [12,13,14]. In addition, several medical and surgical methods of sand removal have been tested and published [12,13]. A reduced frequency of enterolithiasis has been reported for Californian horses without daily access to pasture grazing [30]. The only quantification of sand in pathologic cases [14] found from 0.1 up to 1.6 g of sand per 100 g of stool, which is clearly less than observed in our experiment. Although sandy soils seem to be linked to sand ingestion, soil ingestion can also occur on a wide range of soil textures [1], when geophagy was reported in 13 Australian horse farms, on soils with textures ranging from heavy clay to light sandy soil. It is noticeable that none of these studies quantified the soil ingestion. Another problem linked to soil ingestion can be associated with the telluric bacterium *Rhodoccus equii*, which can affect foals health [31]. Even if the aerial route of exposure is responsible for the most severe pathology, such as pneumopathy, most foals probably acquire a subclinical infection via the alimentary route [32]. Moreover, necrotizing enterocolitis can be observed. These three examples demonstrate that soil ingestion by horses can be an important parameter affecting their health.

### 4.4. Perspectives

This first study showed clearly that soil ingestion by grazing horses is not negligible at 500 g per day, which may affect animal welfare and health. More detailed work is necessary to precisely quantify the effects on soil ingestion such as different grazing managements, differences between horse categories (as for instance draft horses vs. sport ponies) or in horse farming specific conditions, such as raising in paddocks or in boxes with access during only some hours to a bare ground area. Such data would be very helpful to reduce the incidence of health problems related to ingested soil.

## 5. Conclusions

The first study to evaluate the ingestion of soil showed that grazing sport horses would ingest 3 to 4% of soil corresponding to approximately 600 g of dry soil per horse per day, which is quite similar to reported soil ingestion in adult cattle grazing in good conditions. These amounts are not negligible in the context where different pathologies due to ingested soil and sand are described and show the need to limit this involuntary intake in order to ensure the welfare of grazing horses. The amount of herbage offered and the sward height appear to be useful tools to limit soil ingestion as previously shown in other herbivorous animals.

## Figures and Tables

**Table 1 animals-11-02109-t001:** Ingestion of grass and soil depending on the Daily Herbage Allowance (DHA, % of the body weight), period and DHA × period interaction.

Variable	DHA Level			Residual Variation ^1^		*p*-Value		
	2%	3%	4%	RSD	RSE	DHA	Period	DHA × Period
Grazing								
Pre-grazing sward height (cm)	12.0	12.2	11.5	-	0.9	NS	-	-
Post-grazing sward height (cm)	3.1 ^bB^	4.1 ^abA^	4.4 ^aA^	-	0.7	*	-	-
Grass AIA ^2^ content (g/kg DM ^3^)	17.2	18.9	19.2	-	4.2	NS	-	-
Grass intake (kg DM/d)	14.0	15.2	14.4	1.0	-	NS	NS	NS
Body weight (kg)	610 ^b^	630 ^a^	628 ^a^	7.3	-	**	NS	NS
Faeces								
Faecal output (kg DM/d)	7.5	8.0	7.7	0.2	-	NS	NS	NS
DM digestibility of ingested grass	0.466	0.476	0.470	0.01	-	NS	*	NS
DM digestibility of total diet	0.470	0.480	0.473	0.01	-	NS	*	NS
Faecal AIA content (g/kg DM)	60.3 ^a^	55.3 ^ab^	51.1 ^b^	4.5	-	*	**	NS
Soil AIA content (g/kg dry soil)	697 ^A^	678 ^B^	697 ^A^	-	13.9	*	-	-
Daily Soil Ingestion								
SIR ^4^ (% of total DM ingested)	4.5 ^a^	4.1 ^ab^	3.7 ^b^	0.4	-	*	**	NS
Absolute QIS ^5^ (g/horse)	648 ^a^	624 ^a^	543 ^b^	54.3	-	*	***	NS
Relative QIS (g/100 kg BW)	107 ^a^	99 ^ab^	89 ^b^	9.5	-	*	***	NS
Relative QIS (g/kg metabolic BW)	5.3 ^aA^	5.0 ^abA^	4.4 ^bB^	0.5	-	*	***	NS

Significance of *p*-Values for tested effects was <0.05 for *, <0.01 for **, <0.001 for *** and >0.05 for NS (non-significant); ^a,b^ Values within a row with different superscripts (lowercase letters) differ significantly at *p* < 0.05; ^A,B^ Values within a row with different superscripts (capital letters) differ significantly at *p* < 0.1; ^1^ Mixed models on individual data gave a residual standard deviation (RSD) but simple models on plot data (two animals grazing together) resulted in residual standard error (RSE); ^2^ AIA: acid insoluble ash; ^3^ DM: dry matter; ^4^ SIR: soil ingestion rate; ^5^ QIS: quantity of ingested soil.

## Data Availability

All data is available from the authors and can be accessed after a simple request containing a reasonable motivation.

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
