# Peer review of "Ingestion of Soil by Grazing Sport Horses"

_animals, 2021, doi:10.3390/ani11072109_

Round 1

Reviewer 1 Report

This is a very interesting study. Certainly, the results depend on many approximations, but the methodology is replicable and therefore the results are likely to be good approximations. Are there studies in other species that demonstrate this (regarding the ingested dry matter and therefore soil as well as for the amount of faeces)?

Please add the daily amount of soil that remains in the horses and discuss how plausible the result is. According to my calculations it is between 100 and 200 g. What happens with all the soil in the gastrointestinal system of the horses? How is it eliminated?

There were found differences between the first grazing round and the following ones. Please comment on the reason/s that led to lower AIA contents in the faeces and the soil ingestion rates during the first experimental period compared to periods 2 and 3.

Details

DM and CP should be written in full the first time they are used.

Line 169: The sentence “….DHA tended to be (3.1 cm for 2% vs. 4.1 cm for 3%, P<0.10). “ is incomplete.

It is confusing that for the quantity of ingested soil two acronyms are used: QIS and IQS. Use one of them only.

Author Response

Dear reviewer,

please find in the attached document our answers (in red) under each of your remarks (in black).

yours,

Stefan Jurjanz

Reviewer 2 Report

This study aims to evaluate ingestion of soil in horses on varying levels of forage pasture. The authors find that as the grass becomes shorter, soil ingestion escalates, potentially placing horses at risk of gastrointestinal disease such as sand colic. The manuscript is clearly laid out, and well designed. I have the following points for the authors to consider:

Major

1.    The authors should consider the role of ingestion of soil content and the risk of enterolith formation (see ‘Evaluation of dietary and management risk factors for enterolithiasis among horses in California’ by Hassel et al.
2.    For equine veterinarians reading this manuscript, I think it would be helpful to define ‘sward height.’ This is ultimately one of the most important points of the manuscript, and the terminology may not be familiar to all readers

Minor

1.    Misspelling of Kyrgyzstan (line 47)
2.    I am not certain of the meaning of the sentence ‘soil can carry an important microbiologic charge in the stomach’ (line 50). Please clarify

Author Response

Dear reviewer,

please find in the attached document our answers (in red) to your remarks (in black).

yours sincerely,

Stefan jurjanz
